# Hybrid Network–Spatial Clustering for Optimizing 5G Mobile Networks

**Aristotelis Margaris \*, Ioannis Filippas and Kostas Tsagkaris**

Incelligent PC, 17121 Athens, Greece; yf@incelligent.net (I.F.); kt@incelligent.net (K.T.)
**\*** Correspondence: am@incelligent.net

**Abstract:** 5G is the new generation of 3GPP-based cellular communications that provides remarkable connectivity capabilities and extreme network performance to mobile network operators and cellular users worldwide. The rollout process of a new capacity layer (cell) on top of the existing previous cellular technologies is a complex process that requires time and manual effort from radio planning-engineering teams and parameter optimization teams. When it comes to optimum configuration of the 5G gNB cell parameters, the maximization of achieved coverage (RSRP) and quality (SINR) of the served mobile terminals are of high importance for achieving the very high data transmission rates expected in 5G. This process strongly relies on network measurements that can be even more insightful when mobile terminal localization information is present. This information can be generated by modern algorithmic techniques that act on the cellular network signaling measurements. Configuration algorithms can then use these measurements combined with location information to optimize various cell deployment parameters such as cell azimuth. Furthermore, data-driven approaches are shown in the literature to outperform traditional, model-based algorithms as they can automate the optimization of parameters while specializing in the characteristics of each individual geographical zone. In the context of the above, in this paper, we tested the automated network reconfiguration schemes based on unsupervised learning and applied statistics for cell azimuth steering. We compared network metric clustering and geospatial clustering to be used as our baseline algorithms that are based on K-means with the proposed scheme—hybrid network and spatial clustering based on hierarchical DBSCAN. Each of these algorithms used data generated by an initial scenario to produce cell re-configuration actions and their performance was then evaluated on a validated simulation platform to capture the impact of each set of gNB reconfiguration actions. Our performance evaluation methodology was based on statistical distribution analysis for RSRP and SINR metrics for the reference scenario as well as for each reconfiguration scheme. It is shown that while both baseline algorithms improved the overall performance of the network, the proposed hybrid network–spatial scheme greatly outperformed them in all statistical criteria that were evaluated, making it a better candidate for the optimization of 5G capacity layers in modern urban environments.

**Keywords:** 5G networks; clustering; location information; network measurements; planning; re-configuration; coverage; quality; optimization

## 1. Introduction

Cellular networks provide both high capacity and geographical coverage wireless networking to mobile users worldwide. Mobile network operators use multiple layers of these networks as a way to utilize both the earliest generations (e.g., GSM and UMTS) as coverage layers (i.e., to provide coverage and high availability) and high-performance newer generations (e.g., LTE, NR) as capacity layers. Capacity layers are designed to follow the traffic density providing the highest possible data rates and lowest latency, which are key to supporting demanding Internet services. To maximize the effectiveness of the capacity layers, the cellular network planning [1–3] process needs to take into consideration

the localized mobile user characteristics of each different zone. This specialization process provides the best possible coverage with the lowest interference, resulting in high quality of service. Radio planning teams traditionally use an iterative method of re-configuring cell parameters to maximize the performance of these cell layers. During this process, network measurements are being collected that are then used to assist the parameter benchmarking. These measurements are either performed by specialized hardware operated by radio engineers, or they can be based on probes and real mobile terminals that operate in test modes. The generated datasets consist of network records that contain signaling information related to network operations and actions such as channel quality measurements and network-related information from upper network layers.

The inclusion of location information in these measurements is of key importance for the identification of coverage deficiencies and optimum network reconfiguration [4–8]. To acquire the location information, various techniques [9–11] have been shown to estimate the mobile terminal's location within a specific zone by using only the radio channel quality measurements performed by the mobile terminal devices during handover-related operations. Areas such as university campuses, city malls, and entertainment venues (that are generally placed right outside the urban zones) have very complex population density and traffic demand patterns related to other more rural zones. These factors are causing the static planning techniques to underperform and the need has risen for more advanced analytical solutions that can leverage mobile terminal location information. This information has great benefits as it improves the accuracy of the detection of quality degradation metrics to a specific point in the map (e.g., a traffic hot-zone).

Analysis of the measurement data can also be conducted with automated methods [12] to generate reconfiguration actions that will change the layout of the network, further improving the radio coverage and quality and increasing the capacity layer's performance. In this paper, we analyzed the literature of model-based and data-driven algorithms for cell reconfiguration, and focused on unsupervised learning [13–16] methodologies that have proven their effectiveness in a large number of network-related optimization tasks. Traditional approaches such as the clustering of network KPIs has been shown to correctly detect and segment the network elements into meaningful groups, which greatly reduces the processing overhead and is very robust to the noisy nature of network data. The inclusion of location information in measurement datasets makes algorithms that perform spatial clustering a key asset to the discovery of underlying human and traffic hot-zones.

In this research, we are proposing a novel algorithmic scheme that combines the merits of both the network KPI and the location data clustering techniques. This approach is shown to correctly identify the problematic zones of the underlying area and propose cell azimuth reconfiguration actions in a completely automated manner. The algorithm uses the spatial component to identify clusters of human densities in the zone and then applies filtering on the centroid values of each cluster to narrow down to true traffic hot-zones (i.e., where user terminal activity is high). To quantify the performance benefits of the proposed scheme against the baseline approaches, system-level simulation of an indicative 5G network deployment was performed. The simulation also included a validated, complex mobility model that will increase the realism of the simulation and better showcase the effects of each reconfiguration scheme. Analysis of the experimental results produced in this research validates the hypothesis that the proposed algorithm outperforms the baseline approaches and can be an important component of an automated radio access network planning tool. Table 1 lists all abbreviations used in this document, along with the term they describe.

The rest of this paper is organized as follows. State-of-the-art analysis and related work related work are presented in Section 2. Section 3 introduces the problem statement of the current work. In Section 4, there is an extended description of the solution methodology and the different methods implemented for the experiments. Section 5 presents the performance evaluation procedure, which includes a detailed overview of the simulation-based evaluation platform as well as detailed performance evaluation results for each algorithm,

along with the statistical distribution analysis of network KPIs and the comparison of improvements compared to the initial scenario. Section 6 presents a number of discussion topics based on this work. Finally, the paper is concluded in Section 7.

**Table 1.** Table of notations used in this paper.

| Notation | Description |
| --- | --- |
| 3GPP | 3$^{rd}$ Generation Partnership Project |
| gNB | Next Generation Node B (NR) |
| eNodeB | Enhanced Node B (LTE) |
| RSRP | Reference Signal Received Power |
| SINR | Signal-to-Inference-plus-Noise Ratio |
| DBSCAN | Density-based Spatial Clustering of Applications with Noise |
| GSM | Global System for Mobile Communications |
| UMTS | Universal Mobile Telecommunications System |
| LTE | Long Term Evolution |
| NR | New Radio |
| KPI | Key Performance Indicator |
| SON | Self-Organizing Network |
| D2D | Device-to-Device |
| MIMO | Multiple-Input and Multiple-Output |
| Lat, Lng | Latitude, Longitude |
| RAN | Radio Access Network |
| OSS | Operations Support Systems |
| DU, UR, SU, RU | Dense Urban, Urban, Sub-Urban, Rural |
| ISD | Inter-Site Distance |
| PCA | Principal Component Analysis |
| UE | User Equipment |
| CDF | Cumulative Distribution Function |

## 2. Related Work

Recent research activity has demonstrated a plethora of solutions for self-organizing networks as next generation architectures in 3GPP standards. SON targets in enhancing network performance KPIs such as capacity and quality of service. 3GPP Release 8 classified SON into three main categories: self-configuration, self-optimization, and self-healing. Machine learning based approaches, both supervised and unsupervised, have been proposed for all three categories of use cases. Some of the most widely used techniques include artificial neural networks for configuration optimization, demand prediction, or resource allocation [12]. Unsupervised techniques that have been tested in the context of SON include self-organized maps, game theory, hidden Markov models, and of course, clustering [13,14]. Clustering-based solutions appear in various use cases. In [15], a user-centric clustering algorithm was used to minimize the high load on cells. In [16], the K-means algorithm was used to group users to define spatial beams and increase system capacity. In caching applications, clustering is applied to data users to determine the set of the most influential users, the content of which will be cached and re-used in Device-to-Device (D2D) communication [8]. Another class of approaches uses clustering on the 5G cells to define groups with increased mobility and load activity to apply optimization actions in a smaller number of cells such as in [17]. While user-centric unsupervised approaches have been proposed in the context of SON [8,15,16], the literature that applies unsupervised techniques in the planning and reconfiguration of 5G networks is not very extensive. Thus, we extend the aforementioned ideas in the context of optimization of the azimuth steering of the 5G gNB cells in mobile networks. This work is the first attempt, to the best of our knowledge, to apply a user-centric clustering-based solution to this aim.

Different clustering algorithms have been heavily explored and compared in the literature. K-means is considered the most typical example of partitioning approaches and has been widely used in numerous applications. Its main disadvantage is that it requires the number of clusters K as an input and results in spherical shaped clusters. It also assumes

that variables have similar variance, which is not difficult to ascertain in the exploratory analysis steps. It usually converges in a smaller number of iterations than most other clustering algorithms, but requires a comparison of the average silhouette coefficient for different numbers of parameter K, which is time consuming [18].

Another class of clustering algorithms that have gained in popularity in the past decade is the density-based approaches. The most indicative algorithm in this class is DBSCAN, which regards clusters as dense areas of data points that are separated by less dense areas. Density-based approaches have a major advantage over partition-based ones, which is the ability to discover clusters of arbitrary shape. However, DBSCAN and other algorithms in this class have difficulties in finding clusters with varied densities, as they use a global density threshold for the formation of clusters [19]. To overcome this weakness, many variants of the initial DBSCAN algorithm have been proposed such a hierarchical DBSCAN [20], which will be employed in this work. The idea behind HDBSCAN is to produce many DBSCAN outcomes through increasing density thresholds. A dendrogram is built based on these outcomes to yield the hierarchical cluster structure. To decide the final clustering, it extracts a set of "significant" clusters at different levels of this dendrogram. HDBSCAN was chosen over other alternatives as it is performance efficient, easy to configure, and has not been used in the context of the optimization of 5G networks, to the best of our knowledge.

## 3. Problem Statement

The optimization process of 5G capacity layers [2–7,21] involves the identification of optimum values for the adjustable cell/site parameters listed in Figure 1. Each of these parameters contributes in a different way to the network quality and has its own constraints, which will be analyzed in this chapter to detect the most suitable candidate for an automated reconfiguration scheme. Site location (incl. height) is of key importance for the topology of the 5G network as it will dictate the actual position of the cells that will cover an area.

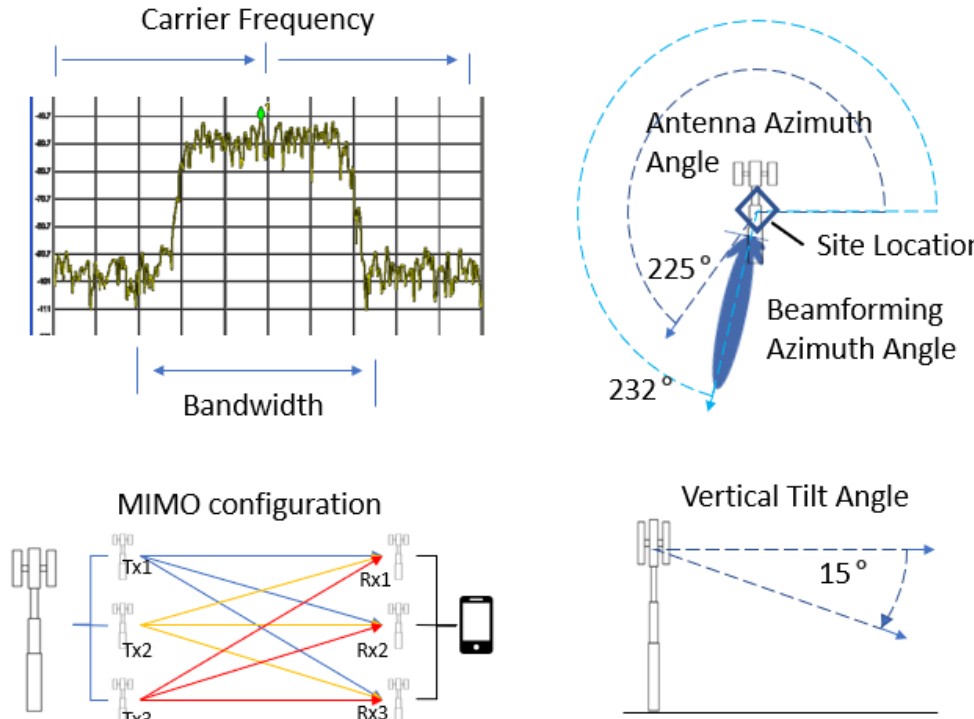

**Figure 1.** Adjustable 5G cell/site parameters.

The cell site usually includes a number of cells (typically 3–5) of single or multiple radio access technologies, and it greatly affects the coverage area of each cell. It is generally analyzed combined with the azimuth angle, mechanical or electrical vertical tilt, which are all key to estimating the achieved coverage for each included cell. The site locations are placed in areas with clear line of sight and with an increased height. They also require computation units for the radio processing, which needs specialized housing structure and power supplies. This makes the relocation, addition, and maintenance of new site locations a complicated and costly process. Cell azimuth angle describes the direction in which the receivers of the radio signal will acquire the maximum antenna gain. This will result in higher data rates, provided that the interference levels allow for transmission. The azimuth angle is an installation parameter on the site that is changed in a relatively slow pace, typically after reconfiguration and optimization orders from the radio optimization team. Cell azimuth steering can play a key role in improving the coverage and quality of users within a zone where hot-zones have been formed due to urban structures such as plazas, shopping malls, recreation areas, and university campuses. In addition to the physical cell azimuth configuration, beamforming azimuth steering is also available via MIMO configuration of the antenna. However, the value range of this change is restricted, usually in a $+/-20$ degree range from the physical azimuth angle, making the physical cell azimuth a more impactful planning parameter. Cell vertical tilt (electrical or mechanical) can also be an important configuration parameter of the 5G cells. This parameter also affects the center of maximum antenna gain, which we described in the azimuth steering case as they are important factors in the 2-D antenna gain computation. In modern deployments of 5G, however, the quality is due to the impact of pathloss and absorption for the carrier frequency. However, the use of higher carrier frequency bands minimizes the expected range of the cells, making for denser networks with higher vertical tilt values. Another parameter that can also be considered is the radio bandwidth and carrier frequency of the cell. These parameters can greatly impact the maximum capacity of the cell as well as the achieved radio and are not flexible parameters to configure, mostly due to the competition-based spectrum acquisition process that is conducted by the regulation authorities. The 5G access network MIMO/spatial multiplexing operational mode is also a tunable aspect of the cellular network. Depending on the relative position of the mobile terminal with respect to the gNB cell, its radio capabilities, and the current (instantaneous) channel conditions, multiple radio access paths can be utilized to multiply the capacity of the radio link by a significant factor. While MIMO configuration optimization is a very active topic of research in this field, data-driven approaches have been shown to perform better in real time optimization and are not in the scope of network planning. Out of these configuration actions, the cell azimuth steering is expected to provide the greatest benefits for the cases of dense urban traffic hot-zones that will be part of our study. It can be configured to multiple values and does not include extreme cost or other blocking factors, apart from the manual labor required for its adjustment. Therefore, for the rest of this study, we focus on algorithmic schemes that will reconfigure the physical azimuth steering of cells within a target area based on mixed location and network metric measurements.

## 4. Solution Methodology

This paper focused on automated techniques that maximize the coverage and quality of service (RSRP, SINR, and per user downlink throughput) of modern 5G capacity layers using optimum azimuth steering. The high-level concept and context of our solution when implemented in a mobile operators' network is shown in high level in Figure 2 and can be briefly described as follows.

The cell network was analyzed with respect to its coverage and capacity needs and with respect to its traffic intensity in different areas. A set of cells was identified as the 'candidate' to be reconfigured in terms of their azimuth in order to cover these coverage and capacity needs. Measurements from these cells are collected in the form of radio access network (RAN) KPIs including RSRP, SINR, and throughput in software systems that

are traditionally responsible for the operation and management of RANs, namely, RAN operations support systems (OSS). Location information is also collected either through the OSS or through third party geo-location systems. Algorithms responsible for parameter configuration and optimization then use these data in order to derive parameter settings (e.g., azimuth angles). These algorithms are typically hosted in an intelligent (data-driven) optimization server, as depicted in Figure 2, which can be either a sub-part of a RAN OSS or a separate software application. The derived azimuth settings are then applied to the selected cell(s) and the newly produced KPIs are expected to increase. The above should be a continuous and (as much as possible) automated process to ensure that the network always operates with the optimal parameters in all different contexts (i.e., time and space variations).

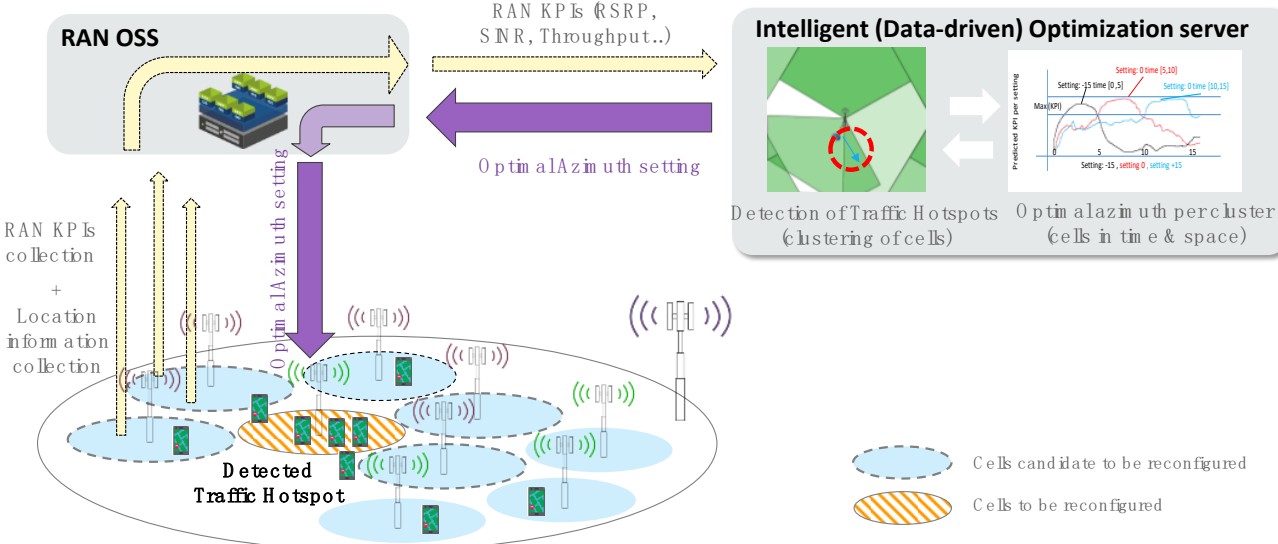

**Figure 2.** High level concept and context of our optimal azimuth steering solution.

In this paper, we opted to evaluate three data-driven techniques as part of the intelligent optimization server in Figure 2, namely, two from the literature and one proposed scheme. The selected techniques include a combination of unsupervised learning and statistical analysis, which have been shown to have a robust performance in diverse geographical, temporal, and operational contexts. Independently of its algorithmic specificities, each technique was evaluated by following a set of general steps as follows: (a) an initial measurement phase of the network in an initial configuration; (b) processing of the measured dataset to derive insights for the underlying cellular network; (c) conversion of these insights into reconfiguration parameters; (d) application of the reconfiguration on the network; and (e) evaluation of the impact of the actions with respect to the initial scenario. These are described in detail in the next sub-sections, followed by an algorithm complexity analysis section.

### 4.1. Baseline Algorithm 1—Network Metrics Clustering

A rich literature [22–25] of unsupervised techniques for network data analysis exists with applications in multiple optimizations, planning, and reconfiguration tasks related to cellular networks. This process involves the acquisition of a dataset that includes network metric measurements that will then be used to derive clusters (i.e., groups) of data that present similar behavior. These groups are then linked to insights by applying the cluster label analysis process, which gives characteristics to these groups that can then be included in algorithms for optimization. In the case of azimuth angle optimization, clustering of network metrics can be used to determine a set of cells labelled as, for example, "cells of low coverage quality and high traffic demand". This is an indicator that a reconfiguration action should be performed in the specific cell and/or its neighboring cells that will improve the

radio quality. Based on the approaches found in [22–24], appropriately tuned K-means execution followed by statistical label traceback analysis can robustly provide automated results for the select network metrics. In addition, we can monitor the performance of the clustering operation by inspection of the cluster boundaries in various projected spaces (e.g., with the principal component analysis (PCA) method). After the clustering operation, we translated the cluster label to actions of cell azimuth steering. For this purpose, location information was indirectly derived by each cell sector's geometry—its assumed coverage—and the algorithm indicates that the population density hot-zone is spatially correlated to this area. However, the actual position of the hot-zone cannot be accurately pinpointed, so this algorithm focuses on moving neighboring cells toward the center of the problematic cell, which is arguably a drawback of this algorithm. The algorithmic process of this method is as follows: Having clustered the UEs on the available network features (RSRP and SINR values) each cell is characterized by the KPI scores of the clusters that are serving. When a "bad" cell is found (low score for both KPIs), the derived action is to rotate its closest "good" cell to the direction of the geometric center of the area covered by the "bad" cell to improve their RSRP and SINR scores. The method can be seen in Baseline Algorithm 1.

---

**Baseline Algorithm 1:** Azimuth Steering based on Network Clustering

---

**Output**: Azimuth Steering Action for each gNB

**Input**:

1. Serving gNB for each UE
2. Cluster label for each UE
3. RSRP score (low/medium/high) for each UE
4. SINR score (low/medium/high) for each UE

**Begin**

Cells <- Set of Serving gNBs

Clusters <- Set of Cluster labels

**Foreach** Cluster in Clusters do:

1. Cluster RSRP score <- majority RSRP of Cluster's UEs
2. Cluster SINR score <- majority RSRP score of Cluster's UEs

**End**

**Foreach** *Cell* in *Cells* do:

1. Find the Subset of Clusters that are served by *Cell*
2. Assign to *Cell* RSRP and SINR scores based on scores of this subset
3. Calculate the distances from other *Cells*

**End**

**Foreach** *Cell* in Low Score Cells do:

1. Calculate the geometric center of the *Cell* coverage
2. Find the closest high-score cell to *Cell*
3. Calculate the delta azimuth between the high-score cell direction and the geometric center
4. Rotate high-score cell to make the delta azimuth zero

**End**

**End**

---

### 4.2. Baseline Algorithm 2—Spatial Clustering

Spatial clustering is the process of applying unsupervised learning to identify clusters of geospatial significance (density). The K-means algorithm [22–24] that was also included in Baseline Algorithm 1 has also been shown to be an effective approach for the detection of spatial centroids. The algorithm requires measurements that include location information (in the form of latitude, longitude), user equipment device identifier, and temporal information (in the form of timestamp). The K-means algorithm generates centroids that are population densities detected throughout the underlying area. By identifying population density centers, we can then make decisions as to what azimuth steering action we can take to move the azimuth angle of the antennas toward these centers, maximizing the mobile terminal's RSRP. This approach can be considered as an improvement vs. the symmetrical, hexacomb-based initial scenario and the rest of the literature, however, it does not come without its own drawbacks. Population density and traffic hot-zones cannot be differentiated by this method due to the lack of network metric information in the clustering process. This results in all density centers being included in the azimuth steering process, which can result in maximizing coverage for zones that are of very low traffic, making minimal impact in the improvement of the network operation. For the spatial scenario, UEs are clustered only on their spatial coordinates. Again, each gNB is associated to a subset of the clusters and is rotated to center its direction to the geometric center of these clusters. The steps of the spatial clustering algorithm can be shown in Baseline Algorithm 2.

---

**Baseline Algorithm 2:** Azimuth Steering based on Spatial Clustering

---

**Output**: Azimuth Steering Action for each gNB

**Input**:

1. Serving gNB for each UE
2. Cluster label for each UE
3. RSRP measurement for each UE
4. SINR measurement for each UE

**Begin**

**Foreach** *Cell* in *Cells* **do**:

1. Find the Subset of Clusters that are served by *Cell*
2. Calculate the geometric center of the clusters centroids in that subset
3. Calculate the delta azimuth between the *Cell* direction and the geometric center
4. Rotate *Cell* to make the delta azimuth zero

**End**

**End**

---

### 4.3. The Proposed Scheme: Hybrid Network—Spatial Metric Clustering

The proposed scheme is an approach that aims at exploiting the techniques of both approaches above-mentioned (Baseline 1 and 2). It is using the spatial clusters as a method of detecting the exact location of the high population density zones that is key to the correct computation of azimuth steering actions while using the network metric cluster traceback analysis to distinguish between zones with high traffic demand and idle user zones. This essentially differentiates between clusters that are traffic hot-zones and population hot-zones, which is the key improvement vs. the Baseline 2 case. The accuracy of the azimuth

steering, achieved by the inclusion of the location information, is also expected to make the proposed scheme outperform the Baseline 1 case, which generalizes the location of the traffic hot-zone within the cell's coverage area. Due to the nature of the data distributions of the spatial and non-spatial, the literature indicates that hierarchical DBSCAN (HDBSCAN) [26] is the most suitable clustering approach to accurately segment the data, since it is non-parametric, easy to configure, efficient in training time, and able to discover clusters of varied densities. In particular, this algorithm will detect data clusters in each dimension separately, which will allow for a 2-step identification of location centers and deficient cell quality centers. For the computation of the azimuth delta, the same computation as in Baseline 2 was applied on the part of the centroid that contains the position of the high traffic cluster. In summary, in the hybrid approach, the network and spatial methods are combined. First, UEs are clustered on their spatial coordinates along with their network measurements. Then, similar to the network approach, each gNB is characterized by the network scores of its associated cluster. Finally, when a "bad" cell is found, it is rotated to center its direction with the geometric center of its associated clusters such as in the spatial method.

---

**Proposed Algorithm 3:** Azimuth Steering based on Network—Spatial Clustering

---

**Output**: Azimuth Steering Action for each gNB

**Input**:

1. Serving gNB for each UE
2. Cluster label for each UE
3. RSRP score (low/medium/high) for each UE
4. SINR score (low/medium/high) for each UE

**Begin**

  *Cells* <- Set of Serving gNBs

*Clusters* <- Set of Cluster labels

**Foreach** *Cluster* in *Clusters* **do**:

1. Cluster RSRP score <- majority RSRP of *Cluster's* UEs
2. Cluster SINR score <- majority RSRP score of *Cluster's* UEs

**End**

**Foreach** *Cell* in *Cells* **do**:

1. Find the Subset of Clusters with low SINR and low RSRP scores that are served by *Cell*
2. Calculate the geometric center of the clusters' centroids in that subset
3. Calculate the delta azimuth between the *Cell* direction and the geometric center
4. Rotate *Cell* to make the delta azimuth zero

**End**

**End**

---

### 4.4. Complexity Analysis

The use of HDBSCAN [20,25] offers the advantage that the number of clusters does not have to be configured such as all density-based clustering algorithms. Another great property of the algorithm is that when the feature space of the training data is not very big

such as in our case, its time complexity is O(*nlogn*), which is sub-quadratic and very competitive to K-means. In our experiments, grid search included approximately ~100 parameter configurations and each configuration lasted less than 2 min for the most cases, with the whole procedure finishing in a few hours. It should be noted that the most time-consuming configurations were the ones with high epsilon values (cluster_selection_epsilon), so these should be treated with caution.

## 5. Performance Evaluation

System-level simulators compatible with the latest 4G and 5G radio access modules have been used throughout the literature for testing and validation of various cellular network optimization schemes. They are also used by radio planning and optimization engineers during the process of manual parameter reconfiguration. We performed our study in a validated 4G/5G cellular network simulator for both the initial reference scenario and the optimized configurations generated by each algorithm. We designed the simulation scenario to include a number of traffic hot-zones that can be found in areas such as a shopping mall placed in a dense urban zone. This type of cellular network zone presents with complex mobility patterns due to the different areas of points of interest within the venue. The evaluation process will start with measurements in the initial scenario for the establishment of the network performance baseline, then for each algorithm, we will perform the execution based on the measurement data, which will result in new scenarios that will include the reconfiguration. We will then perform separate simulation executions generating new network measurements for each reconfiguration scheme. The results will be analyzed by means of statistical distribution analysis on coverage (RSRP) and quality (SINR and user downlink throughput) to measure the benefits of each algorithm and compare them. We will also comment on the expected vs. actual results based on each algorithm's design principles in conjunction with the selected scenario.

### 5.1. Simulation Platform Overview and Scenario

The simulation software used for this study was a 5G cellular network simulator [22,27] based on 3GPP specifications [28–30] that can be used for macro-simulation of network functionalities and user mobility scenarios of 4G and 5G cellular networks. The radio channel component supports multiple cellular network elements such as LTE eNodeB, Pico cells, and NR gNB. It is based on standard-based propagation models specified in [28,29] for various urban environments. Its radio quality model translates the link quality (SINR) to the achieved throughput, which is used to serve the generated traffic of the mobile terminals. It also includes a network planning module that was used to automatically generate the initial hexacomb topology. It can easily apply the reconfiguration actions of the selected algorithms to create new measurements based on the impact of the produced azimuth reconfiguration. The simulator also includes a realistic user mobile terminal mobility module that can be parametrized to form hot-zone population density patterns based on the behavioral movement model, as shown in [22,27]. This model's parameters include predefined areas of interest (denoted by geometrical area definition, average dwell time, and probability of arrival) as well as specific paths that are selected based on statistical distributions. The simulator network traffic module was based on [31] and the traffic level configuration was tuned to follow the modern traffic demand levels as shown in Table 2. For the baseline scenario in this study, we opted to analyze a cellular network area during its transition from an existing 4G cellular network to a 5G capacity layer. This deployment typically follows layered installations of gNB cells on identical site and azimuth configuration with its predecessor technology. The hexacomb placement scheme is the most common geometrical pattern and is often used as the initial topology. The simulation includes a small DU area (0.11 km$^2$) with a high density of users (3635 mobile/km$^2$) as well as a shopping mall zone with multiple indoor zones. The dense cell deployment requires a minimum of a 176 m inter-site distance, resulting in six gNB sites with three

sector antennas Table 2. Standard traffic demand parameters were selected following the simulation scenarios used in existing 5G benchmarking studies.

**Table 2.** Simulation scenario parameters.

| Parameter | Simulation Value |
|---|---|
| Playground Dimensions | 340 m × 340 m |
| Area (meters$^2$) | 0.11 km$^2$ |
| Mobile Terminals | 400 |
| Population Density | 3635 mobile/km$^2$ |
| Total Traffic Density | 1.32 Gbps/km$^2$ |
| Per Mobile Traffic rate | 36.25 Mbps/mobile |
| # of gNB sites/cells | 6 (18 gNB sectors) |
| Inter-Site-Distance | 176 m (DU) |
| gNB Vertical Tilt | 26° |
| gNB carrier and band | 2.1 GHz (2 × 15 MHz) |
| Radio Propagation Model | Uma 5G |

The mobility model of the simulation [22,27] was parametrized with a set of areas of interest (shops, restaurants, parking lots, cinemas etc.) that are relevant to shopping malls Figure 3. Each area was linked with predefined paths according to the actual topology of the underlying area for added realism. The areas of interest were split into various categories characterized by mobility information and traffic usage levels. Areas that generate notable mobile user hot-zones are recreation areas (high mobile usage and high dwell time), which are typically coffee shops, restaurants, etc. and offline areas where high density and high dwell time was observed but mobile phone usage was limited (e.g., parking lots and cinemas).

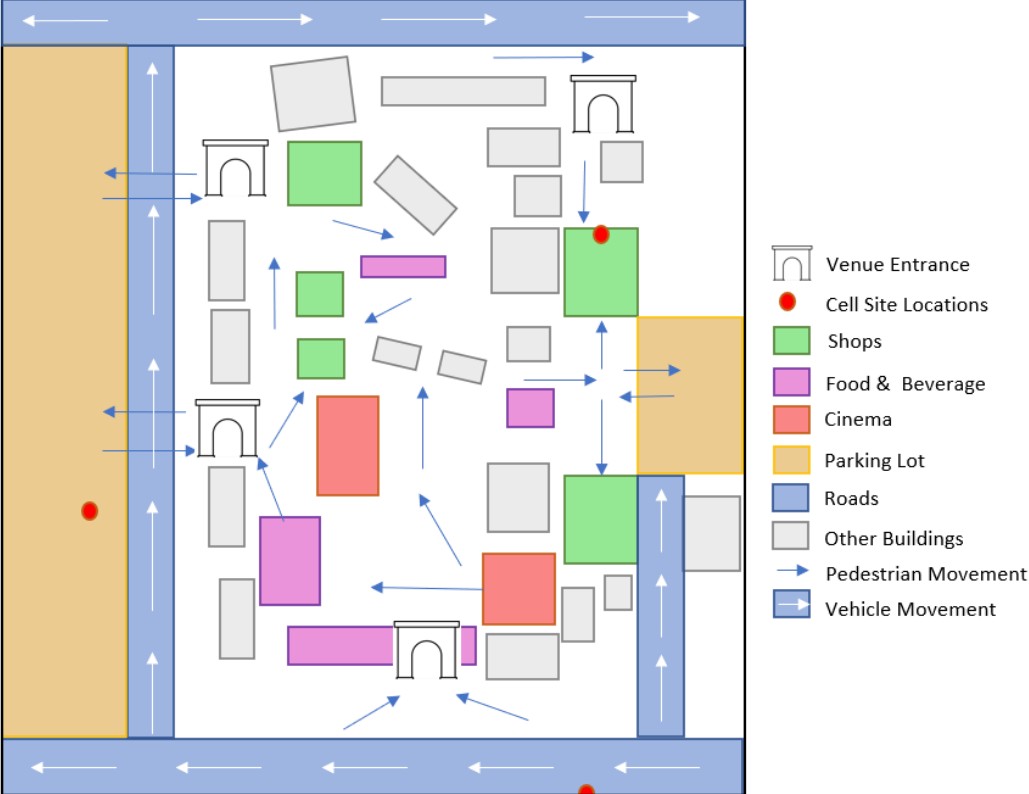

**Figure 3.** Specified areas of interest and example paths with respect to the site location in the existing topology.

### 5.2. Simulation Measurement Dataset Analysis

In Table 3, we can see the dataset structure of the mobile terminal simulation measurements that were generated for each test case. It contains fields ranging from spatial to temporal data, IDs of the related mobile user equipment as well as the active gNB cell and the core network metrics that are related to coverage (RSRP, active RSRP), quality (SINR), and achieved data rates (throughput). It must be noted that the sampling frequency of such datasets is closely related to the rate of change that the spatial and network metric parameters have. In typical cases, such datasets have measurement intervals in the order of seconds or even lower, therefore producing a very large amount of data within a small period of time. This is something that can affect each algorithm's performance and should also be considered during the design of automated reconfiguration schemes.

**Table 3.** Mobile Terminal simulation measurement dataset analysis.

| Field | Description | Example Values |
|---|---|---|
| Mobile ID | An identifier for each mobile terminal included in the measurements | Mob_1,2 ... N |
| Timestamp | Time identifier of the measurement | 1638035033 |
| Serving Cell | An identifier of the cell that the mobile terminal is connected to | Cell 2_1, Cell 3_2 |
| RSRP | Received signal strength from serving gNB (idle) | −72 dBm |
| Active RSRP | Received signal strength from serving gNB (transmission) | −72 dBm |
| SINR | Signal-to-Interference-and-Noise Ratio (transmission) | 9 dB |
| Downlink Throughput | The data rate achieved by the mobile terminal during download | 34.2 Mbps |
| Latitude | The latitude position of the mobile terminal | 25.61 (deg) |
| Longitude | The longitude position of the mobile terminal | 37.61 (deg) |

### 5.3. Network Metric Clustering Analysis

The network metric clustering aims at the separation of the measurement data into groups with similar behavior based on the network metrics included (RSRP, active RSRP, SINR, downlink throughput). The clustering effectiveness can be seen in the projection of the multi-dimensional dataset in a projected plane such as the PCA projection. In the 2-dimensional component space of PCA, we used coloring to define each cluster detected, expecting clear distinction between the clusters [Figure 4]. We are also generating centroid classes to profile each cluster with a label that can be used to detect measurement groups of high or low qualities such as the ones shown in Table 4. We could see that there were 10 generated clusters with meaningful partitioning that uses a combination of SINR and RSRP cluster labels to provide clear network performance information. For instance, Cluster 0 is described as "Medium SINR, Medium RSRP" and Cluster 1 as "High SINR, Very Low RSRP", etc., as seen in the overview of the cluster qualitative results. Conditional probability analysis was used in the centroids to identify serving cells that are more related to the clusters with the deteriorated coverage and quality. Provided the list of problematic cells, we used their neighboring cells to improve the radio quality of the cell sector's area, therefore producing azimuth steering computations.

**Table 4.** Clustering result legend including color, index, and SINR/RSRP profile quality class.

| Color | Cluster Index | SINR Profile | RSRP Profile |
|---|---|---|---|
| ● | 0 | Medium | Medium |
| ● | 1 | High | Very Low |
| ● | 2 | High | High |
| ● | 3 | Low | Very Low |
| ● | 4 | Very Low | Medium |
| ● | 5 | Very High | Very High |
| ● | 6 | Low | Low |
| ● | 7 | Very Low | Very Low |
| ● | 8 | Low | High |
| ● | 9 | High | High |

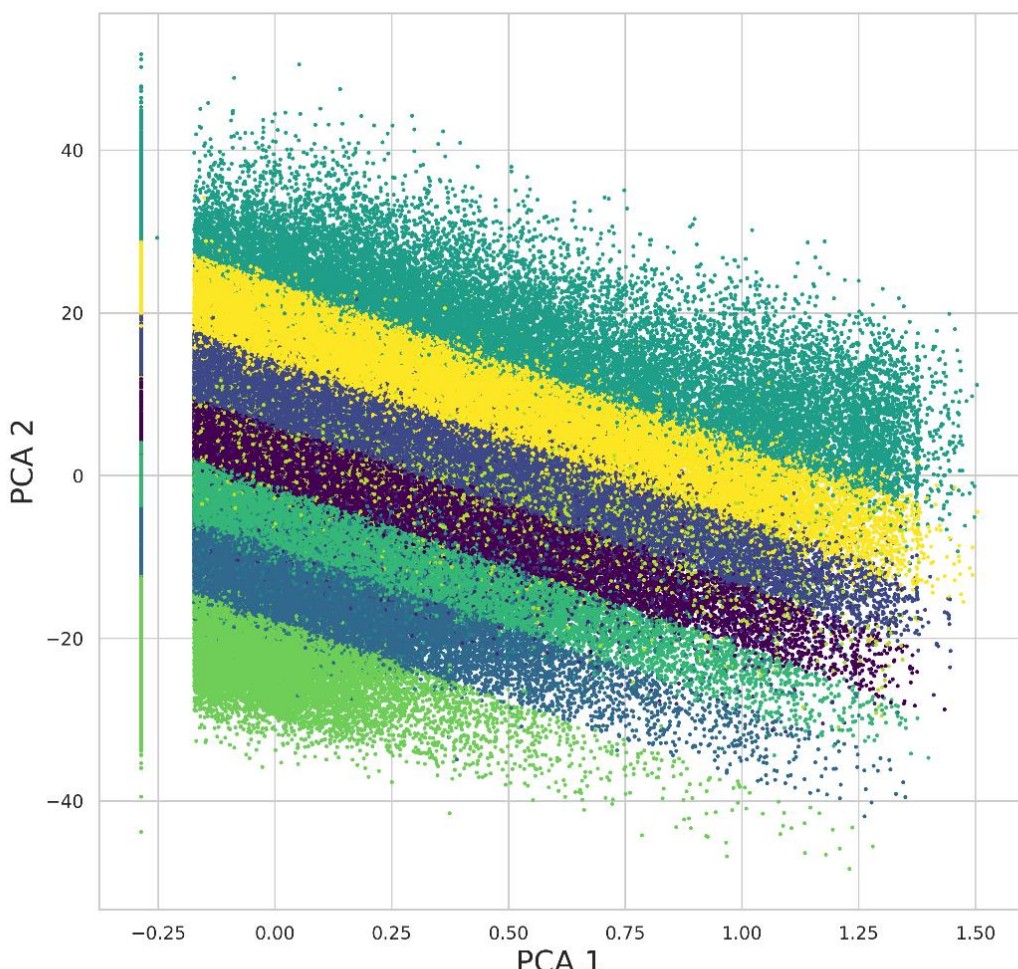

**Figure 4.** Network metric clustering results (PCA) colored by cluster.

### 5.4. Spatial Clustering Analysis

The spatial clustering algorithm is based on the automated detection of population density hot-zones. It leverages the localization information acquired from the dataset by transforming it into population density centroids that can be used to compute azimuth steering, which will increase the number of mobile terminals that are close to the center of the sector antenna. As seen in Figure 5, the algorithm correctly identified densities that were correlated with the predefined zones of the mobility model parametrization area. This hints that the performance of the generated azimuth steering actions is an improvement with respect to the symmetrical deployment. However, the resulting population density areas did not differentiate on the traffic demand of the mobile terminals. We expect that this will result in azimuth steering actions toward low traffic zones that will be considered as incorrect.

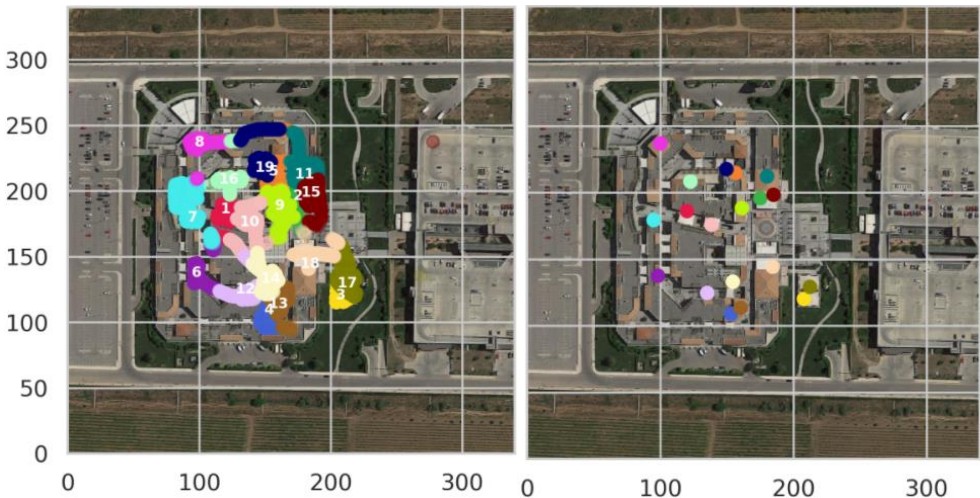

**Figure 5.** Spatial clustering generated zones (**left**) and centers (**right**).

### 5.5. Hybrid Network–Spatial Clustering Analysis

The hybrid network–spatial clustering's goal is to simultaneously detect clusters with deteriorated radio quality and signal strength as well as identify their exact location using the location centroid components. Based on this, we expect it to produce the best possible azimuth steering actions that will maximize the effectiveness of the gNB's antennas. The results of the algorithm execution are shown in Figure 6. They include both location and clear indication of the network metrics for each centroid. Using this information, the algorithm filters the centers into a smaller set that only includes actual traffic hot-zones.

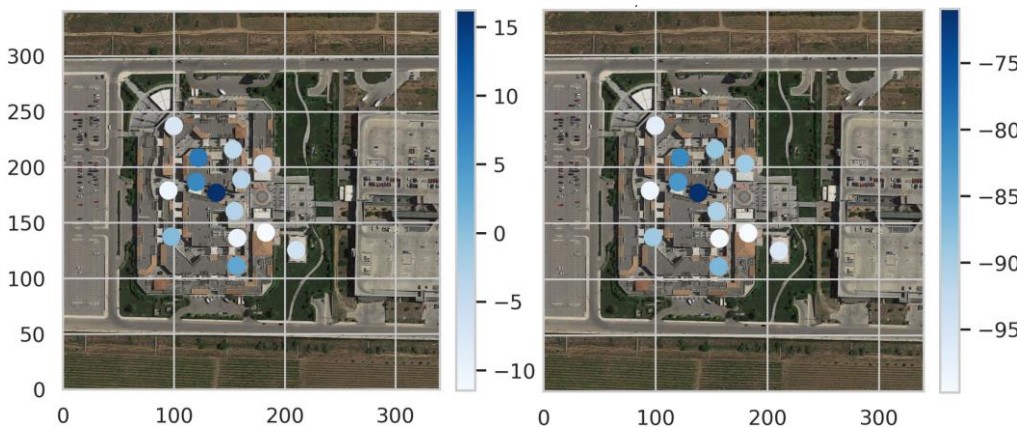

**Figure 6.** Hybrid network–spatial centroids colored by their SINR (**left**) and RSRP (**right**) metrics.

### 5.6. Algorithm Performance Evaluation

The goal of each algorithm in this benchmarking is the maximization of coverage (RSRP) and quality (SINR) of the total cellular network area. In addition, we will evaluate the achieved, per-device downlink throughput to monitor the impact of the changes in the achieved quality of service. For this purpose, we used the statistical cumulative distribution functions of these KPIs as well as the 25[th], 50[th], and 75[th] percentile to acquire quantitative results on the improvements with respect to the reference scenario. To evaluate the distributions [2,3,6], we are looking for right shifts toward higher RSRP/SINR/normalized per user downlink throughput values, which would indicate network metric improvement. In Figure 7, it is evident that the RSRP distribution shifted to the right from 2–5 dB in various ranges of the distribution for all of the optimization schemes. The spatial and network KPI clustering seemed to improve the reference scenario by ~1 dB, which is acceptable,

however, the hybrid clustering approach improved by more than 2.5 dB, making it the best candidate solution for the coverage optimization goal.

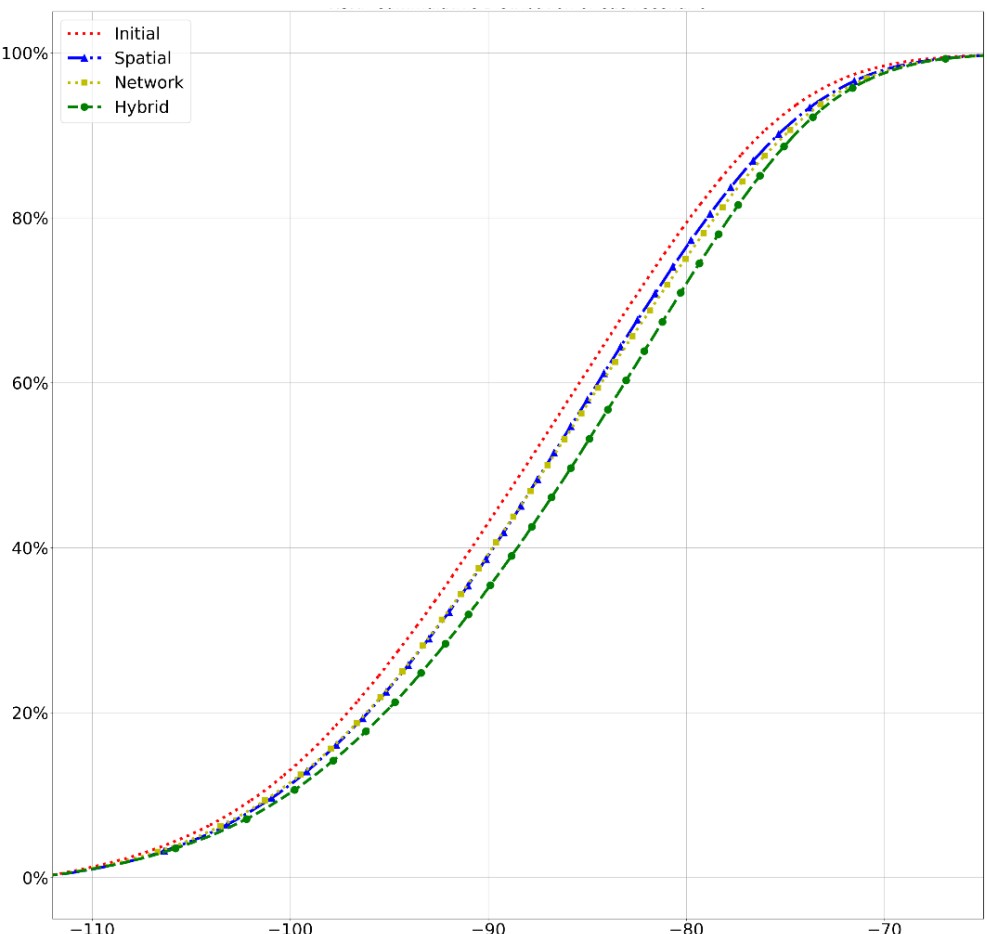

**Figure 7.** Performance evaluation of each algorithm performance vs. the initial scenario for RSRP (dBm) CDF.

The same distribution analysis was then performed on the SINR [Figure 8] measured by the mobile terminals during their transmission. RSRP gains alone are not adequate measures of quality improvement, mostly due to the possibility of increased interference caused by overlapping antenna sectors. In the analyzed data, we could see that the SINR had also improved by all algorithm schemes, therefore indicating that the radio quality had not deteriorated. The proposed scheme's distribution was still a clear improvement vs. the two baseline algorithms by a factor of 2–4 dBs, proving that it can correctly capture the interference constraints of the network as well as specialize to the locations of high traffic density. The same benefits of the proposed scheme are also visible in the statistical analysis of the normalized throughput distribution, as seen in Figure 9. It is clear that higher normalized throughput ratio values receive more measurement density in the case of the proposed scheme.

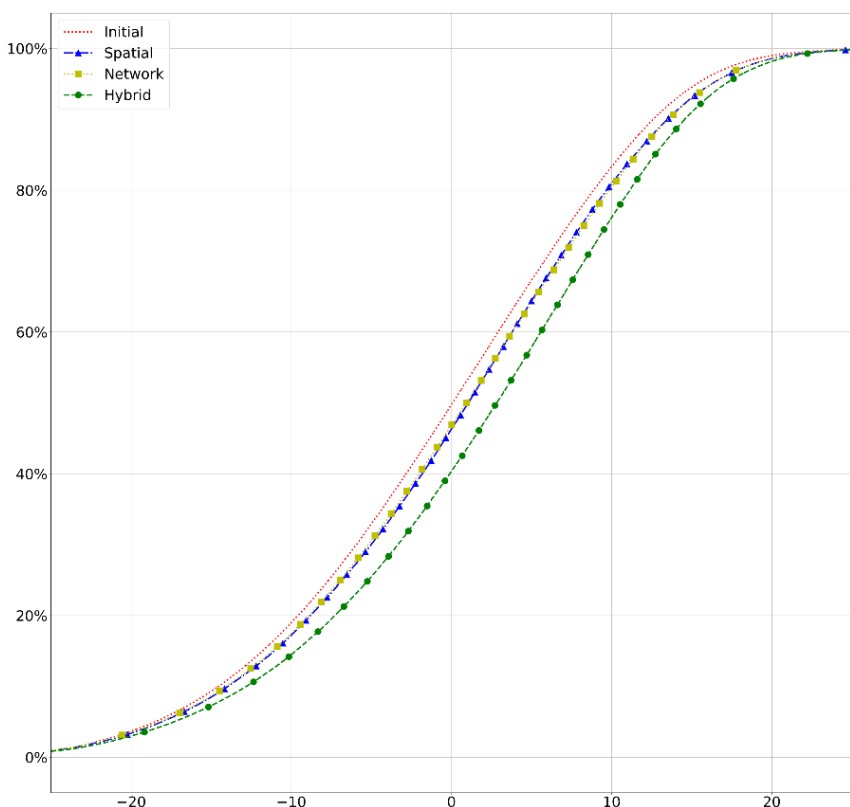

**Figure 8.** Performance evaluation of each algorithm vs. the initial scenario for SINR (dB) CDF.

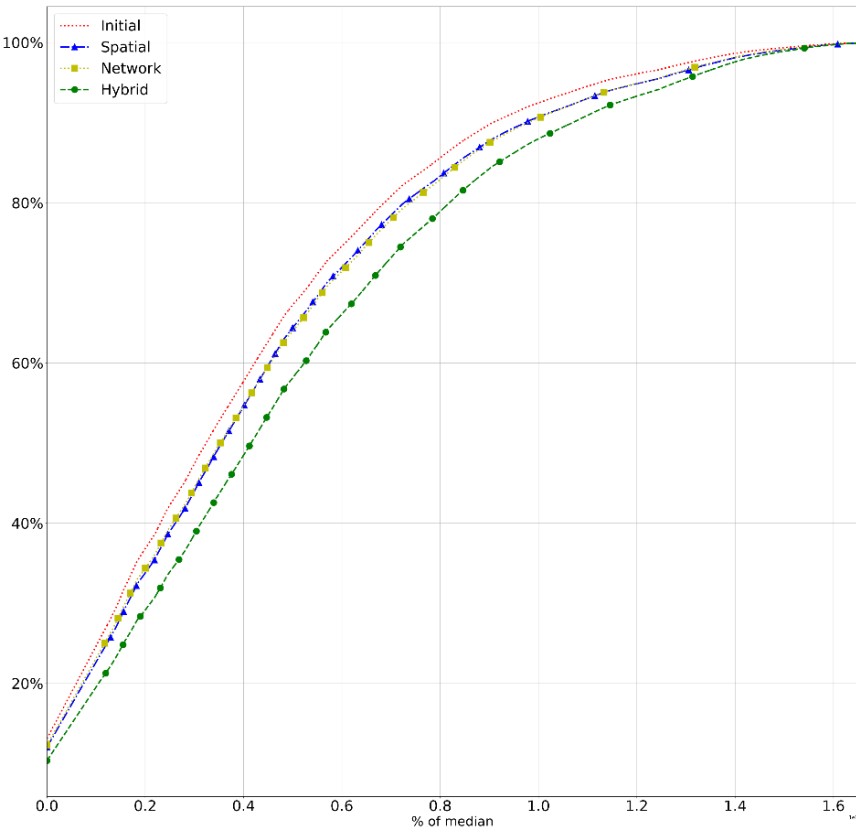

**Figure 9.** Performance evaluation of each algorithm vs. the initial scenario for normalized throughput (ratio vs median) CDF.

To summarize the comparison between the algorithms, we computed the distribution percentiles of the data generated by all the different reconfiguration schemes for all evaluation KPIs, RSRP gains Figure 10, SINR gains Figure 11, and normalized per user downlink throughput Figure 12. The proposed scheme–hybrid network spatial clustering outperformed the baseline algorithms for this type of environment by a factor of more than ~100% in most cases. Another notable result was that the spatial and network algorithms, although they performed completely different reconfiguration actions, showcased very close performance and both their overall statistical distribution as well as the percentiles improved the network performance to a similar degree.

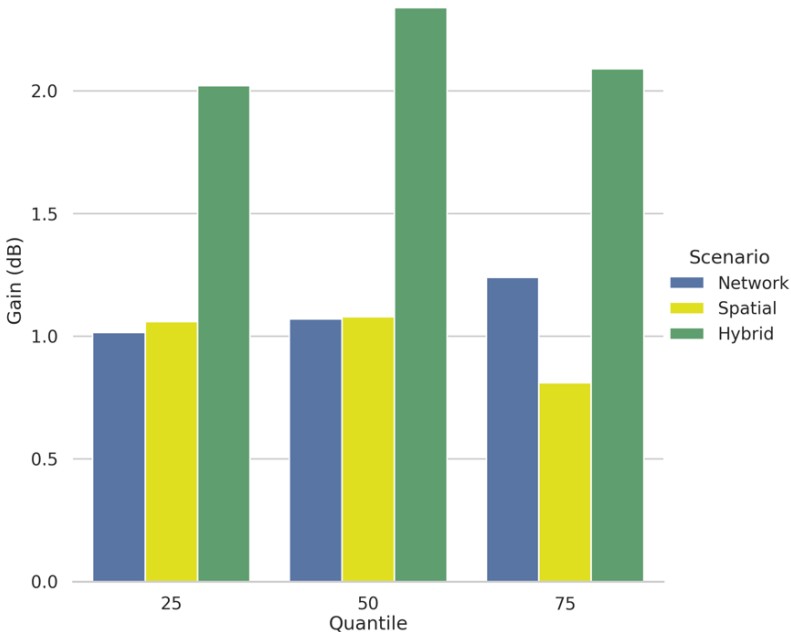

**Figure 10.** Comparison between the different algorithms' improvement vs. the baseline for RSRP gain (dB).

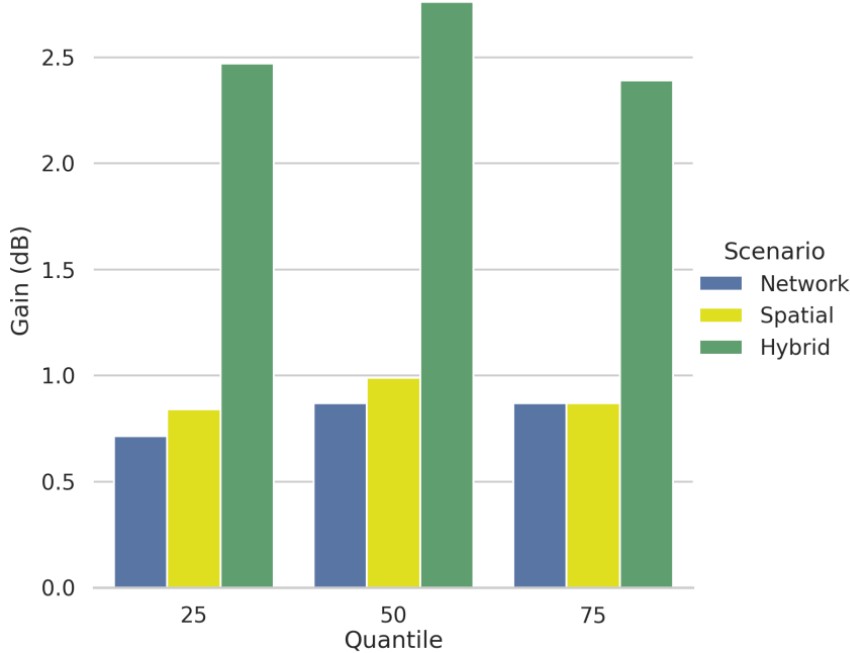

**Figure 11.** Comparison between the different algorithms' improvement vs. the baseline for SINR gain (dB).

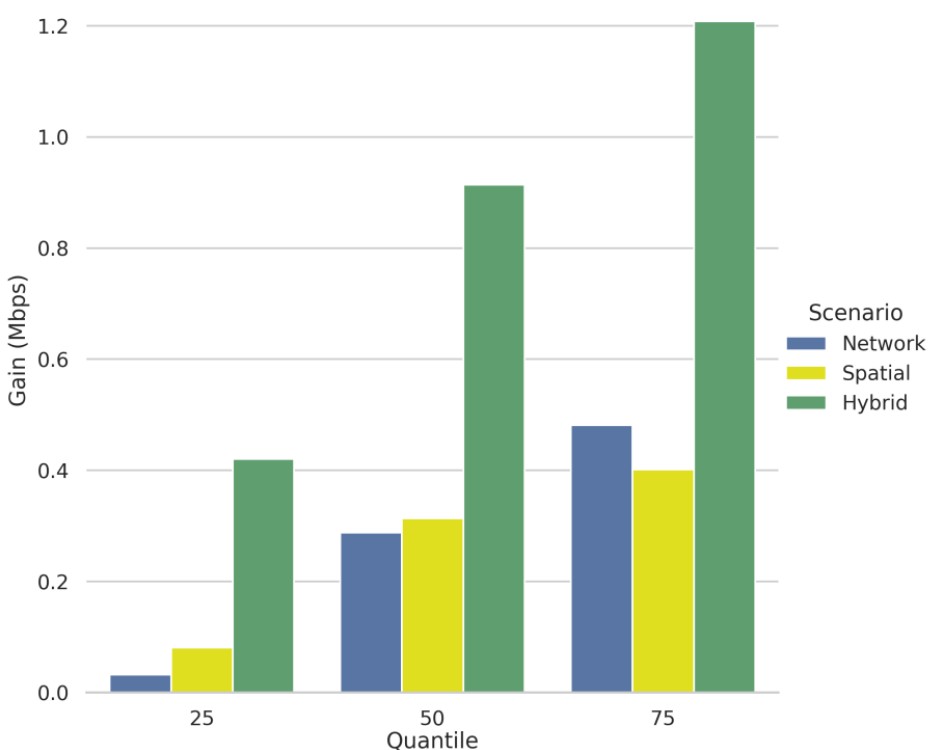

**Figure 12.** Comparison between the different algorithms' improvement vs. the baseline for per user normalized throughput gain (Mbps).

## 6. Discussion

Optimum 5G capacity layer deployment involves a large number of parameters to be considered and tuned, however, azimuth steering can provide significant improvement in coverage and quality with the minimum amount of effort and cost. The azimuth steering problem is shown to be very closely related to the geometry of the active users of the underlying network. The directional gain of the sector antenna (or patch antennas for gNB) provides significant improvement in RSRP, SINR, and achieved user throughput, even in non-line of site propagation (which is very highly present in modern environments). In this study, we focused our efforts on a case of 5G deployment for the 2.1 GHz band, however, it is in the scope of the continuous technology rollout that millimeter-wave carrier frequencies will be utilized, as seen in various studies [1]. Radio transmission at higher frequencies tends to benefit less from wave spreading phenomena such as refraction in comparison to the core cellular frequencies (0.5–3.5 GHz range). Millimeter-wave transmission is very sensitive to cell antenna direction and line of sight plays a critical role on the quality of the link. It will be interesting to expand this research toward applying the proposed schemes on millimeter and even micro-wave carrier experimental deployments and compare the benefits with respect to this study.

Traditional model-based approaches for optimum configuration can be shown to be ineffective in areas with very high concentration of mobile phone users, therefore, we believe that data-driven approaches are key to optimized deployments. Automated reconfiguration algorithms are effective if used with accurate measurements of the network location information of the mobile terminals. It greatly reduces the time and effort required from processes such as continuous manual analysis and reconfiguration performed by teams of radio engineers. Simulation software is also a possible alternative that can be used to replace real measurements, if careful considerations and scenario design have been made, which in turn would require expert knowledge on telecommunications and user behavioral analysis in various urban contexts. Automated azimuth steering methodologies are shown to effectively utilize unsupervised learning components such as the K-means,

X-means, DBSCAN, and other variants. These algorithms generate robust results that are a good fit to the statistical distributions of network data measurements. Clustering label traceback analysis is also an important component, as shown in the implementation of Baseline 1 and the proposed algorithm. It is based on conditional probability analysis to create network metric labels that provide meaningful insights for the cells.

## 7. Conclusions

In this paper, we designed and tested data-driven methodologies that allow for automated coverage and performance maximization of the 5G capacity layer. This approach is suited for areas that include hot-zone population densities and network traffic patterns mostly found in dense urban and urban zones. The algorithms consist of a combination of unsupervised learning, statistical analysis, and analytical coverage optimization, based on measurements performed on mobile terminals from an initial, hexacomb-based symmetrical network deployment. These measurements include temporal, network, and location information, which are utilized differently by each included algorithm. The outputs of these algorithms are azimuth angle changes in various cells of the underlying network that maximize the RSRP, SINR, and per user downlink throughput. Specialized simulation software for 5G radio simulation was used as the validation platform of these algorithms with realistic mobility patterns and radio environments that are found in dense urban shopping mall zones. All algorithms generated acceptable results that improved the network coverage and quality in relation to the reference scenario. The two baseline algorithms underperformed by a significant factor compared to the proposed scheme in both metrics due to their lack of information to correctly specialize to the zone's characteristics. In particular, the network metric clustering approach (Baseline 1) can only use the sector's geometry to estimate the traffic hotspot location, which leads to incorrect actions. Baseline 2 algorithm, on the other hand, utilizes the location information to identify population density hot-zones. However, without the usage network metric information, this can lead to azimuth steering toward inactive population centers. The proposed scheme's combined spatial and network approach is designed to further refine the Baseline 2 algorithm using network metric inputs in a manner similar to Baseline 1. It was shown that this combined approach is the most appropriate candidate for the automated azimuth steering reconfiguration of 5G capacity cell layers.

**Author Contributions:** Methodology, I.F.; writing—original draft preparation, A.M.; writing—review and editing, K.T. All authors have read and agreed to the published version of the manuscript.

**Funding:** This work has been supported in part by Incelligent PC and by the European Union's Horizon 2020 research and innovation program under grant agreement no. 871249 (Research and Innovation Action), LOCalization and analytics on-demand embedded in the 5G ecosystem for Ubiquitous vertical applicationS (LOCUS).

**Institutional Review Board Statement:** Not applicable.

**Informed Consent Statement:** Not applicable.

**Data Availability Statement:** Not applicable.

**Acknowledgments:** This work has been supported in part by Incelligent PC and by the European Union's Horizon 2020 research and innovation program under grant agreement no. 871249 (Research and Innovation Action), LOCalization and analytics on-demand embedded in the 5G ecosystem for Ubiquitous vertical applicationS (LOCUS).

**Conflicts of Interest:** The authors declare no conflict of interest.

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
