# Peer review of "Hybrid Network–Spatial Clustering for Optimizing 5G Mobile Networks"

_applsci, doi:10.3390/app12031203_

Round 1

Reviewer 1 Report

This paper develops a hybrid network-spatial clustering for optimizing 5G mobile networks. Overall, it is interesting. I have some comments for the authors to consider.

-- To me, the related work is not properly reviewed. Other than briefly listing the existing works in Sec.2, I would suggest showing some basic discussions on how these works relate to the current one. Such discussions shall also highlight the motivations of the current work. 

-- It can be better to use a figure to illustrate the key cell parameters in Table 1, as some of them may have ambiguous meanings. For example, azimuth angle may also refer to the beam scanning/forming angle.  

-- lines 259, page 8: ... is the best..., please add conditions for such statement. 

-- after reading sec.4, I am not sure about the paper's innovations. It looks like several existing algorithms are described. Please justify. 

-- Fig. 9: it is better to differentiate curves by line styles not colors.

-- Sec.6: the authors are invited to provide some discussions on network optimisations in millimetre-wave frequency bands 

Author Response

Dear reviewer,

We greatly appreciate the feedback that you have composed for our submitted paper, and we focused all our efforts into improvements based on your suggestions. In detail:

Q1: To me, the related work is not properly reviewed. Other than briefly listing the existing works in Sec.2, I would suggest showing some basic discussions on how these works relate to the current one. Such discussions shall also highlight the motivations of the current work.

A1: The 'related work' section has been reformed to include more details about the research process that was followed and led us into the selected algorithms for the reference problem.

Q2: It can be better to use a figure to illustrate the key cell parameters in Table 1, as some of them may have ambiguous meanings. For example, azimuth angle may also refer to the beam scanning/forming angle. 

A2: We have generated an appropriate figure, illustrating the various configuration options of the gNB cell making a clear distinction between azimuth steering and the other parameters

Q3: lines 259, page 8: ... is the best..., please add conditions for such statement.

A3: Full rephrasing of this section has clarified the justification of the algorithm selection for our use case.

Q4: after reading sec.4, I am not sure about the paper's innovations. It looks like several existing algorithms are described. Please justify.

A4: This section has been revised to clearly explain the thought process behind the state-of-the-art analysis as well as the innovations of the proposed algorithm

Q5: Fig. 9: it is better to differentiate curves by line styles not colors.

A5: We have applied the style changes to better showcase the curve differentiation of each result

Q6: Sec.6: the authors are invited to provide some discussions on network optimizations in millimeter-wave frequency bands

A6: The millimeter-wave RAN use cases of 5G are closely related to the azimuth selection topic due to their sensitivity to LOS/NLOS conditions therefore we have expanded on the topic in the discussion section.

We hope that the applied changes have elevated the quality of this paper to its best potential.

With regards,

Dr. Aristotelis Margaris

Reviewer 2 Report

The paper entitled “Hybrid Network-Spatial Clustering for Optimizing 5G Mobile Networks” proposed a hybrid clustering framework for optimizing 5G mobile networks. It focuses on algorithmic schemes that will reconfigure the azimuth steering of cells within a target area based on mixed location and network metric measurements. To achieve that purpose, it uses unsupervised techniques and statistical analysis to maximize the coverage and quality of service (RSRP and SINR) of modern 5G capacity layers using optimum azimuth reconfiguration.

Generally, the topic conducted in this paper is interesting and may gain high interest from readers thanks to the upcoming trends of 5G mobile networks. I have used k-means, DBSCAN as well as HDBSCAN (an improved version of DBSCAN in which it allows varying density clusters instead of using a global epsilon distance as in DBSCAN), I observed that k-means, DBSCAN, and HDBSCAN could produce good performances for the clustering task. In some cases, DBSCAN and HDBSCAN were even better than k-means since they have the ability to remove noises. Thus, the results seem to be reasonable. Before I vote for an acceptance, authors should revise the paper to further improve its quality. My comments are as follows

- In the Introduction, the authors should highlight the main contributions of the paper in both literature and real-world applications. The current text does not show an overview of the framework authors have proposed in this paper/

- In Related work, authors should discuss possible methods that can perform clustering for an unknown number of clusters and provide high interpretability such as hierarchical clustering, several good examples are [https://doi.org/10.3390/app112311122] and [https://doi.org/10.1007/978-981-15-1209-4_1].  From that highlight the main advantages of the proposed framework over other methods.

- In section 3, add a table of notations used in the paper.

- Algorithms 1, 2, and 3 should be Algorithms themselves, authors should remove the captions of Figures from those algorithms. Make the text in each algorithm clearer, copiable.

- Figures 1, 5, 6, 9, and 10 are not in a good shape. Authors should revise these figures to make the text clearer, copiable, and not be broken when zooming out.

- In section 4.2, describe when and where the dataset was collected? If it is from another study, make a reference. In addition, are there any problems such as missing values in the original dataset, briefly describe them.

- Sections 4.3, 4.4, and 4.5 need to be revised to help readers easily follow the differences between modes 1, 2, and hybrid. The authors used the work SotA many times but did not give an explanation for that. Did you mean SotA is state-of-the-art? Those sections are very confusing in terms of notations and explanations.

- [For the whole paper]: carefully proofread the paper to fix all typos in text and notations, as well as grammar mistakes.

Author Response

Dear reviewer,

We greatly appreciate your meaningful comments on our submitted paper and have led us to perform significant changes into the manuscript. Find bellow a detailed analysis of our additions:

Q1: In the Introduction, the authors should highlight the main contributions of the paper in both literature and real-world applications. The current text does not show an overview of the framework authors have proposed in this paper/

A1: We have also received this feedback from additional reviewers, and we have expanded this section to include the full framework of this paper as well as highlight its contributions to the scientific field.

Q2: In Related work, authors should discuss possible methods that can perform clustering for an unknown number of clusters and provide high interpretability such as hierarchical clustering, several good examples are [https://doi.org/10.3390/app112311122] and [https://doi.org/10.1007/978-981-15-1209-4_1].  From that highlight the main advantages of the proposed framework over other methods.

A2: We have found the suggested literature very relevant, and we have included it in our current references list. Model interpretability of the clustering results as well as the optimum selection of number of clusters hyperparameter has also been covered in this section as well.

Q3: In section 3, add a table of notations used in the paper.

A3: Table of notations has been included

Q4: Algorithms 1, 2, and 3 should be Algorithms themselves, authors should remove the captions of Figures from those algorithms. Make the text in each algorithm clearer, copiable.

A4: We have re-created the algorithm diagrams using a different method to embed them into the document to allow for text selection improved rendering and the revised label.

Q5: Figures 1, 5, 6, 9, and 10 are not in a good shape. Authors should revise these figures to make the text clearer, copiable, and not be broken when zooming out.

A5: Figure 1 has been completely redesigned to be simpler, easier to comprehend and have less text which instead has been incorporated in its respective section. The result figures (9,10) have been updated with different colors, gridlines, different line style per case, single legend for all results and increased size for clarity. Figure 6 will be split into a scatter plot figure (left) and an additional table section where we analyze the cluster profiles. Figure 5 will be increased in size and the background / foreground colors will have more contrast to make it clearer.

Q6: In section 4.2, describe when and where the dataset was collected? If it is from another study, make a reference. In addition, are there any problems such as missing values in the original dataset, briefly describe them.

A6: We understand the reason for the ambiguity with respect to the dataset mentioned in 4.2. The dataset that is being analyzed was generated by the initial simulation scenario referenced in section 4.1 and further analyzed in 5.1. It contains no missing values therefore no imputation or post-processing has been made. We have combined the two chapters and reformed our phrasing to make it clearer to the reader.

Q7: Sections 4.3, 4.4, and 4.5 need to be revised to help readers easily follow the differences between modes 1, 2, and hybrid. The authors used the work SotA many times but did not give an explanation for that. Did you mean SotA is state-of-the-art? Those sections are very confusing in terms of notations and explanations.

A7: We have changed the naming of the algorithms from SotA 1,2 to Baseline 1,2 to avoid any confusion to the readers and we have simplified the terms used for the analysis of each algorithm of section 4. The finalized draft will clearly depict the differentiation between the proposed scheme (hybrid) and the baseline approaches covered.

Q7: [For the whole paper]: carefully proofread the paper to fix all typos in text and notations, as well as grammar mistakes.

A8: We have performed a thorough grammar and spelling correction on the revised paper as well as the captions, graphics, and algorithm sections.

We greatly appreciate the effort you have put into this review, and we feel that it has helped us to meet this journal's quality standards.

With regards,

Dr. Aristotelis Margaris

Round 2

Reviewer 1 Report

Thank the authors for responding to my comments. I am happy to recommend the acceptance of the work now. 

Reviewer 2 Report

I have checked this revision. The authors have significantly improved the quality of their manuscript. They have solved all problems raised in the original version. Thus, I would like to vote for an acceptance.